# Sphingosine-1-Phosphate Levels Are Higher in Male Patients with Non-Classic Fabry Disease

**DOI:** 10.3390/jcm11051233

**Published:** 2022-02-24

**Authors:** Wladimir Mauhin, Abdellah Tebani, Damien Amelin, Lenaig Abily-Donval, Foudil Lamari, Jonathan London, Claire Douillard, Bertrand Dussol, Vanessa Leguy-Seguin, Esther Noel, Agathe Masseau, Didier Lacombe, Hélène Maillard, Soumeya Bekri, Olivier Lidove, Olivier Benveniste

**Affiliations:** 1Reference Center for Lysosomal Diseases, Internal Medicine Department, Groupe Hospitalier Diaconesses—Croix Saint Simon, 75020 Paris, France; jlondon@hopital-dcss.org (J.L.); olidove@hopital-dcss.org (O.L.); 2Center of Research in Myology, UMRS 974, Association Institut de Myologie, INSERM, Sorbonne Université, 75013 Paris, France; damien.amelin@upmc.fr (D.A.); olivier.benveniste@aphp.fr (O.B.); 3INSERM Unit 1245, Normandie Universités, 76000 Rouen, France; abdellah.tebani@chu-rouen.fr (A.T.); lenaig.abily-donval@chu-rouen.fr (L.A.-D.); soumeya.bekri@chu-rouen.fr (S.B.); 4Department of Metabolic Biochemistry, Rouen University Hospital, 76000 Rouen, France; 5Department of Neonatal Pediatrics, Intensive Care and Neuropediatrics, Rouen University Hospital, 76000 Rouen, France; 6UF Biochimie des Maladies Neuro-Metaboliques, Service de Biochimie Métabolique, Groupe Hospitalier Pitié-Salpêtrière, AP-HP, 75013 Paris, France; foudil.lamari@aphp.fr; 7Reference Center for Inborn Metabolic Disease, Jeanne de Flandres Hospital, CHU LILLE, 59037 Lille, France; claire.douillard@chru-lille.fr; 8Nephrology Department, Aix Marseille Université et Centre d’Investigation Clinique 1409, INSERM/AMU/AP-HM, 13005 Marseille, France; bertrand.dussol@ap-hm.fr; 9Internal Medicine and Clinical Immunology Department, Francois Mitterrand Hospital, 21000 Dijon, France; vanessa.leguy-seguin@chu-dijon.fr; 10Internal Medicine Department, Strasbourg University Hospital, 67000 Strasbourg, France; esther.noel@chru-strasbourg.fr; 11Internal Medicine Department, Hôtel-Dieu University Hospital, 44000 Nantes, France; agathe.masseau@chu-nantes.fr; 12Medical Genetics Department, CHU de Bordeaux, INSERM U1211, Université de Bordeaux, 33000 Bordeaux, France; didier.lacombe@chu-bordeaux.fr; 13Internal Medicine Department, Huriez Hospital, University of Lille, 59037 Lille, France; helene.maillard@chru-lille.fr; 14Internal Medicine and Clinical Immunology, Reference Center for Neuromuscular Disorders, AP-HP, Groupe Hospitalier Pitié-Salpêtrière, DHUi2B, 75013 Paris, France

**Keywords:** Fabry disease, hypertrophic cardiomyopathy, sphingosine-1-phosphate, fibrosis, migalastat

## Abstract

Fabry disease is an X-linked lysosomal disease in which defects in the alpha-galactosidase A enzyme activity lead to the ubiquitous accumulation of glycosphingolipids. Whereas the classic disease is characterized by neuropathic pain, progressive renal failure, white matter lesions, cerebral stroke, and hypertrophic cardiomyopathy (HCM), the non-classic phenotype, also known as cardiac variant, is almost exclusively characterized by HCM. Circulating sphingosine-1-phosphate (S1P) has controversially been associated with the Fabry cardiomyopathy. We measured serum S1P levels in 41 patients of the FFABRY cohort. S1P levels were higher in patients with a non-classic phenotype compared to those with a classic phenotype (200.3 [189.6–227.9] vs. 169.4 ng/mL [121.1–203.3], *p* = 0.02). In a multivariate logistic regression model, elevated S1P concentration remained statistically associated with the non-classic phenotype (OR = 1.03; *p* < 0.02), and elevated lysoGb3 concentration with the classic phenotype (OR = 0.95; *p* < 0.03). S1P levels were correlated with interventricular septum thickness (r = 0.46; *p* = 0.02). In a logistic regression model including S1P serum levels, phenotype, and age, age remained the only variable significantly associated with the risk of HCM (OR = 1.25; *p* = 0.001). S1P alone was not associated with cardiac hypertrophy but with the cardiac variant. The significantly higher S1P levels in patients with the cardiac variant compared to those with classic Fabry suggest the involvement of distinct pathophysiological pathways in the two phenotypes. S1P dosage could allow the personalization of patient management.

## 1. Introduction

Fabry disease (FD, OMIM #301500) is an X-linked disorder characterized by defects in the alpha-galactosidase A enzyme activity that lead to the ubiquitous accumulation of glycosphingolipids, mainly globotriaosylceramide (Gb3) and globotriaosylsphingosine (lysoGb3). Depending on the alpha-galactosidase A gene (GLA, Xq22.1 300,644) variant, two main phenotypes have been described [1]. The historical classic disease is associated with markedly reduced or absent enzyme activity and a wide spectrum of symptoms including acral neuropathic pain, progressive renal failure, white matter lesions, cerebral stroke, and hypertrophic cardiomyopathy (HCM). The non-classic phenotype is observed in patients with low but detectable enzyme activity and is almost exclusively characterized by cardiomyopathy [1,2]. According to the last expert consensus document on Fabry disease edited by the European Society of Cardiology, Fabry disease could explain up to 1% of unexplained hypertrophy [3]. More than 1000 pathogenic variants of the *GLA* gene have been described [4]. Whereas deletions, frameshifts, and nonsense mutations of the *GLA* gene have been associated with the classic phenotype, the genotype–phenotype correlation is more unobvious with for missense variants [1,5]. 

To date, two different therapeutic options have been validated in Fabry disease. Intravenous enzyme replacement therapy (ERT) with agalsidase alfa (Replagal^®^, Shire-Takeda, Stockholm, Sweden) or agalsidase beta (Fabrazyme^®^, Sanofi-Genzyme, Amsterdam, The Netherlands) has been available since 2000 for all patients with Fabry disease [6,7]. The benefits of ERT have now been accepted, although the disease seems to progress in some patients [8,9]. The development of anti-drug antibodies that occurs preferentially in classic Fabry males, but also inflammation and secondary fibrosis that would occur in all patients, have been suspected to explain the partial failure of enzyme replacement therapy [8,9,10,11]. Since 2016, another validated option consists in a chaperone molecule therapy named migalastat (Galafold^®^, Amicus Therapeutics, Dublin, Ireland) that can only be given to eligible patients defined by an amenable variant of the GLA gene, usually a missense mutation [12]. De facto, migalastat mainly concerns non-classic Fabry patients. In addition, the definition of amenable variants—that is, those associated with an increase of enzymatic alpha-galactosidase A activity in the presence of migalastat—is controversial [13,14].

Plasma globotriaosylsphingosine, also known as lysoGb3, is the deacetylated derivative of Gb3 [15]. Increased levels of plasma lysoGb3 have been observed in Fabry patients compared to healthy controls, higher in males than in females and higher in classic than in non-classic patients [16]. Although the level of plasma lysoGb3 can predict the clinical phenotype, it does not reflect the burden of the disease [16,17]. Hence, no ideal biomarker exists to manage Fabry disease [16,18]. A recent proteomics approach revealed that the levels of some angiogenesis proteins (fibroblast growth factor 2 (FGF2), vascular endothelial growth factor A (VEGFA), vascular endothelial growth factor C (VEGFC)) and of the cytokine interleukin 7 (IL-7) were significantly higher in Fabry patients, independently of lysoGb3 plasma levels [19]. IL-7 concentration was also correlated with residual enzyme activity in non-classic patients, which may reflect a specific pathophysiology in the non-classic phenotype [19]. Plasma sphingosine-1-phosphate (S1P) has been controversially associated with Fabry cardiomyopathy [20,21]. S1P is implicated in cardiac hypertrophy, cardiac fibrosis, and immune regulation [22]. Treatments targeting the S1P signaling pathway are currently used to control multiple sclerosis (FTY720, fingolimod) [23,24] and have shown promising results in hypertension (PF543) [25]. Whereas the existence of a pro-inflammatory condition in Fabry disease has now been accepted [11,26], we aimed to determine whether S1P levels could be of interest in the management of Fabry disease, notably, in relation to cardiovascular involvement. We measured S1P levels in serum from male patients of the French multicenter cohort FFABRY and assessed statistical correlations with clinical data.

## 2. Results

### 2.1. Patients

Forty-one male patients were included. Their clinical and biological characteristics are detailed in Table 1 and in the Appendix A. Non-classic patients were older than classic patients (median 48.1 years [interquartile (IQ) 43.0–59.8] vs. 37.6 [29.5–46.1], *p* = 0.003). 

### 2.2. S1P Levels Are Higher in Non-Classic Patients

LysoGb3 serum levels were available for 36 patients (17 classic and 19 non-classic patients). The median serum S1P concentration was 193.2 ng/mL [IQ: 168.6–219.4]. S1P concentration was not influenced by the delay between blood sampling and serum thawing (Spearman *p* = 0.9), by patient’s treatment (Mann–Whitney: median S1P 200.3 ng/mL in untreated patients vs. 192.8 ng/mL in treated patients; *p* = 0.8), or by the presence of anti-agalsidase antibodies (*p* = 0.6). S1P levels were not correlated with the time to treatment exposure (Spearman *p* = 0.6) and were not different between missense and non-sense GLA variant carriers (*p* = 0.8); S1P levels did not correlate with age (r = 0.2; *p* = 0.2; Figure 1a), creatinine levels (*p* = 0.6), eGFR (*p* = 0.4), or leucocytes count. S1P concentration was higher in non-classic compared to classic patients (200.3 ng/mL [IQ: 189.6–227.9] vs. 169.4 [IQ: 121.1–203.3], *p* = 0.02, Figure 1b). Of notes, three early diagnosed non-classic patients without HCM had elevated S1P levels (268.2, 219.2, and 191.4 ng/mL).

### 2.3. Correlation between S1P Levels and Interventricular Septum Thickness (IST)

#### 2.3.1. S1P Levels Are Correlated with IST in a Univariate Model

S1P levels did correlate with interventricular septum thickness (IST) assessed by echocardiography (r = 0.46; *p* = 0.02; n = 24 with recent available echocardiographic data; Figure 1c). S1P levels were significantly higher in patients with HCM compared to patients without it: 200.3 ng/mL [IQ: 189.6–227.9] vs. 168.6 [IQ: 126.1–203.6]; *p* = 0.04. 

#### 2.3.2. Multivariate Model

Because HCM was more frequent in non-classic patients (Table 1, *p* = 0.003), and HCM prevalence increases with age, we performed a logistic regression model including S1P serum level, phenotype, and age (Table 2). In this multivariate model, age was the only significant variable associated with an increased risk of HCM (OR = 1.33 [95% IC: 1.05–1.69]; *p* < 0.02); S1P was not associated with it.

### 2.4. S1P Levels Remain Associated with Non-Classic Phenotype in a Multivariate Logistic Regression Model

We then assessed whether S1P levels remained associated with the phenotype in a multivariate logistic regression model including age, S1P, treatment status, and lysoGb3 plasma level, which is known to be associated with phenotype and modified with treatment [11]. Elevated levels of serum S1P remained the only variable statistically positively correlated with the non-classic phenotype (OR = 1.05 (95%CI: 1.01–1.10); *p* < 0.03, Table 2), and plasma lysoGb3 the only variable statistically negatively correlated with the non-classic phenotype OR = 0.97 (95%CI: 0.95–0.99; *p* < 0.03), Table 2). It is of note that lysoGb3 plasma levels and S1P serum levels were not correlated (*p* = 0.9).

### 2.5. S1P Level Is Higher in Non-Classic Patients Compared to Classic Patients Sharing the Same Genotype

In accordance with the possible genotype–phenotype heterogeneity, some patients—five pairs and one trio—shared the same genotype but a different phenotype. Although the small number of patients prevented any statistical conclusion, we observed that S1P level was higher in non-classic compared to classic patients in pairs sharing the same genetic variant, i.e., *p*.Phe337Ser: 229.5 vs. 200.98 ng/mL, respectively in non-classic and classic patients; *p.Trp162Cys*: 230.2 vs. 92.3 ng/mL; *p.Arg112Cys*: 219.44 vs. 190.7 ng/mL; c.802-3_802-2del/*p*? 268.1 vs. 125.5; *p.Arg301Gln*: 219.2 vs. 210.2 ng/mL. The only mismatch concerned the *p.Gln283 ** genotype: a renal transplant classic male had a S1P level of 234.5 ng/mL compared to levels of 189.6 and 199.1 ng/mL observed in the two non-classic paired patients. We then observed that among all classic patients, kidney transplant patients had a trend to present higher S1P levels compared to non-transplant patients (median 227.9 vs. 163.6 ng/mL, *p* = 0.1). 

### 2.6. S1P Provides Additional Information to the Proteomic Signature of Fabry Disease

FGF2, VEGF-A, VEGF-C, and IL-7 have recently been described as a proteomic signature of Fabry disease [12]. Plasma levels of FGF2, VEGF-A, VEGF-C, and IL-7 were available for only 24 patients (9 classic and 15 non-classic patients). Considering these 24 patients only, we performed Spearman tests for each protein in both phenotype subgroups and in the whole group. We did not observe any correlation with S1P serum levels, except for a trend between FGF2 and S1P levels in non-classic patients (r = 0.48, *p* = 0.07). To evaluate whether S1P serum level provides additional and complementary information, we performed a principal component analysis (PCA) including S1P, lysoGb3, FGF2, VEGFA, VEGFC, and IL-7 as active variables (Figure 2 and Figure 3). The two first dimensions of the PCA accounting for 62% of the total variance allowed a good clustering of patients depending on their phenotype (Wilks test, *p* = 0.01) and the presence of hypertrophic cardiopathy (Wilks test, *p* < 0.02). In this model, HCM was correlated with S1P and VEGF-C levels. Interestingly, patient #21 who segregated with the HCM group was a 17-year-old classic Fabry boy with an IST measured at 11 mm, which may suggest a pre-symptomatic signature of HCM.

## 3. Discussion

We observed that S1P serum level was particularly increased in non-classic compared to classic Fabry patients. Like Brakch et al., S1P level did correlate with cardiac disease. Nevertheless, the correlation was no longer observed after stratification on clinical phenotype [20]. In the study of Brakch et al., the 17 Fabry patients involved were heterogeneous (9 males with unknown phenotype and 8 females with different cardiac involvement) [20]. The correlations between S1P level and left ventricular mass and intima media thickness could have resulted from patients’ heterogeneity, as we observed for interventricular thickness in our present study. Another study by Mirzaian et al., conducted with classic Fabry patients, did not report an obvious increase in plasma S1P concentration compared to non-Fabry individuals and suggested that the variability observed between Fabry patients and controls could derive from the heterogeneity in collecting, handling, and storing the samples [21]. Both studies used mass spectrometry to assess S1P levels. In our protocol using ELISA, the conditions for sampling, handling, and storing serum were identical for all patients, and we did not observe any influence of the time from sampling to thawing the samples. Hence, we can explain both previous studies that appeared contradictory at a first glance: S1P levels depend on the clinical phenotype in Fabry disease and are particularly elevated in non-classic patients, i.e., patients who are usually diagnosed with Fabry disease because of a hypertrophic cardiomyopathy. This finding is crucial because it suggests that the non-classic phenotype may involve different pathophysiological pathways with respect to the classic one and would not just consist in a disease continuum. Aerts et al. demonstrated the pathogenic role of lysoGb3 on the smooth muscle cells of vessels, with a proliferative effect [15]. It is now accepted that plasma lysoGb3 level is significantly lower in non-classic Fabry patients and even normal in some women [27]. Brakch et al. demonstrated that S1P also has a proliferative effect, and mice treated with S1P developed a Fabry-like hypertrophic cardiomyopathy [20]. The results of these two experiments may explain the development of Fabry cardiopathy if we consider that lysoGb3 and S1P play a major role, respectively, in the classic and the non-classic phenotype. 

Disturbances in S1P levels have been observed in several other lysosomal diseases with unobvious pathophysiological mechanisms [20]. S1P derives from the phosphorylation of sphingosine by the sphingosine kinases SK1 and SK2 [28]. The levels of S1P are also regulated by degradative enzymes such as S1P phosphatase and S1P lyase [29]. S1P has been involved in local vascular inflammation [30]. It has also been implicated in the regulation of the unfolded protein response and of endoplasmic reticulum stress-induced autophagy via the S1P phosphohydrolase-1 (S1PP1) [31]. In fact, S1P mediates a wide range of cellular actions through five different G protein-coupled S1P receptors in humans (S1PR1-5), whose activation depends on the differential concentration gradient of circulatory S1P physiologically existing between compartments [32]. S1PR1 activation is involved in the egress of lymphocytes from lymph nodes and promotes Th-17 pro-inflammatory polarization of immature T lymphocytes, although it has been associated with more limited fibrosis in the liver and lungs [32]. S1PR1 also participates to the vascular endothelial barrier function, induces bradycardia, and has cardioprotective effects after ischemic conditions [32]. S1PR1 is regulated by endocytosis after binding to S1P or by S1PR2 activation [32]. S1PR2 participates in B cell regulation and has pro-fibrotic effects [32]. Whereas Fabry disease appears as a genetic vasculopathy involving inflammatory processes, disorders of autophagy, and organ fibrosis [26], S1P could have a central role in the pathophysiology. Unfortunately, we did not have enough recent cardiac MRI to assess the association between S1P level and cardiac fibrosis. We do not know yet whether the increase in S1P levels participates or is secondary to vasculopathy in patients with non-classic Fabry disease. Cardiac MRI has become an essential tool to assess the severity of Fabry disease [33]. The comparison of cardiac MRI phenotype with S1P levels could bring new insights in the pathophysiology of the disease. However, S1PRs and S1P gradients should also be studied. The unfolded protein response theoretically observed in non-classic variant patients may explain in part the dissociation of S1P levels between classic and non-classic patients [26]. Lastly, although we did not observe a clear relationship between lysoGb3 and S1P levels, it has been suggested that all the sphingoid bases, and so lysoGb3, might act as structural mimics of S1P [34]. It appears therefore essential to further study S1P concentration gradients, the lysoGb3/S1P balance, and the activation of sphingosine kinases and S1PRs in Fabry patients, especially as numerous S1P modulators are available or under development [32]. The major limitation of this study is the heterogeneity of the patients in terms of age and the prevalence of cardiac disease between phenotype groups. However, this limitation is inherent in studies in rare diseases. We also regret the lack of exhaustive recent echocardiographic data, concomitant to S1P sampling, that would have strengthened the power of the analyses. Phenotype classification was performed using the FFABRY algorithm that determines the phenotype depending on the presence of cornea verticillata and the history of acroparesthesia in males [35]. This classification takes in account the genotype–phenotype heterogeneity [35]. The values of the residual enzymatic activity of alpha-galactosidase and of plasma lysoGb3 levels at baseline, before any treatment, could have allowed a better classification. Unfortunately, many of the patients had been treated for years, before plasma lysoGb3 dosage was available. Moreover, the initial determination of alpha-galactosidase enzymatic activity had been carried out in different settings, using different techniques.

S1P alone did not appear sufficient to discriminate patients with cardiomyopathy. Nevertheless, we observed that serum S1P concentration associated with lysoGb3, VEGF-C, VEGF-A, IL-7, and FGF2 levels could differentiate specific clusters of patients according to their phenotype, notably, the presence of HCM. Further studies in different cohorts of patients are needed to validate these results, in particular using mass spectrometry, in order to determine if these six molecules may help stratify the risk to develop the different symptoms of Fabry disease, which could improve the management of presymptomatic Fabry patients.

## 4. Materials and Methods

The multicenter cohort FFABRY prospectively gathers clinical data and biological samples from patients with an enzymatic and/or genetic diagnosis of FD. All the males with available plasma were included. Patients with a heart transplant were excluded. The phenotypes were determined according to the FFABRY score, as described previously [35]. Clinical data were prospectively collected through a standardized online form. FFABRY scores and Mainz severity score index (MSSI) were calculated automatically according to the scoring system established by Mauhin et al. and Whybra et al., respectively [35,36]. HCM was assessed by cardiac magnetic resonance imagery or echocardiography (interventricular septum thickness (IST) > 12 mm). Estimation of the glomerular filtration rate (eGFR) was based on the CKD-EPI equation [37]. Patients referred to as treated underwent enzyme replacement or migalastat therapy.

Blood samples were collected at the time of inclusion and centralized in our research unit. BD VacutainerTM serum tubes with increased silica act clot activator and BD VacutainerTM heparin tubes were used to collect serum and plasma, respectively, before storage at −80 °C.

We used a Human S1P ELISA kit (catalog E-EL-H2583-Elabscience, Houston, TX, USA) to determine serum S1P concentration according to the manufacturer’s instructions. Briefly, 100µL of plasma samples was added in each well for 90 min at 37 °C and then removed, and 100 µL of biotinylated Ab was added, and the mixture was incubated for 1 h at 37 °C. After washing, a horse radish-peroxidase conjugate was added for 30 min at 37 °C before further washing. The substrate reagent was incubated for 15 min at 37 °C, then the reaction was stopped. Optical density was read at 450 nm with a Spark 10M^®^ reader (Tecan Trading AG, Switzerland), and concentrations were determined with a standard curve. 

LysoGb3 concentration was measured in available plasma samples (n = 36) by ultra-performance liquid chromatography coupled to tandem mass spectrometry (UPLC–MS/MS) as previously described [38]. Anti-agalsidase antibodies were screened in all patients as previously described [38]. 

The plasma levels of fibroblast growth factor 2 (FGF2), vascular endothelial growth factor A (VEGF-A), vascular endothelial growth factor C (VEGF-C), and interleukin 7 (IL-7) had been determined previously for some plasma samples [19].

We used non-parametric tests: Kruskal–Wallis (KW) and Mann–Whitney (MW) comparison tests and Spearman correlation test. Variables are described with median [quartile 1–quartile 2]. We used logistic regression with stepwise selection based on *p*-value for discrete variables and Fisher’s exact t test for contingency. The *p* value for the alpha-risk in all tests was 0.05. The EZR plugin version 1.35v [39] packages for the R software and GraphPad Prism 8 were used. Principal components analyses were performed with package FactoMineR and R software version 3.4.0.

## 5. Conclusions

S1P serum level is significantly higher in non-classic compared to classic Fabry disease patients, suggesting the involvement of distinct pathophysiological pathways in the two phenotypes. Despite its role in vascular disease, S1P alone was not associated with HCM. Serum S1P concentration, associated with lysoGb3, VEGFC, VEGFA, IL-7, and FGF2 levels, allowed the clustering of patients according to their phenotype and the presence of hypertrophic cardiopathy. This work opens new perspectives of research in Fabry disease, in particular focused on S1P concentration gradients, lysoGb3/S1P balance, and sphingosine kinases and S1PRs activation.

## Figures and Tables

**Figure 1 jcm-11-01233-f001:**
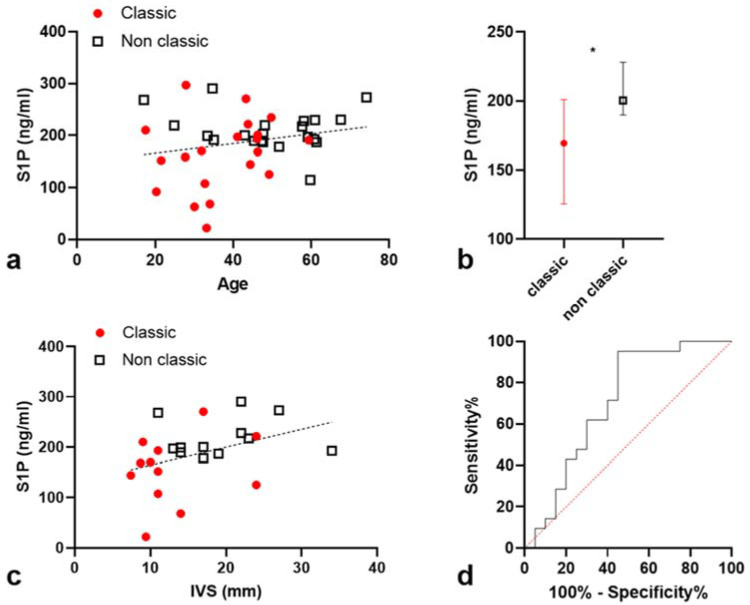
(**a**). Sphingosine -1-Phosphate (S1P) serum levels as a function of age; linear regression. Spearman correlation; *p* value not significant. (**b**). S1P levels in classic and non-classic Fabry males (*p* = 0.024). (**c**). S1P serum levels as a function of interventricular thickness (r = 0.46; *p* = 0.02). (**d**). S1P levels and phenotype. ROC curve area 0.70 ± 0.08 (*p* = 0.02). * *p* < 0.001.

**Figure 2 jcm-11-01233-f002:**
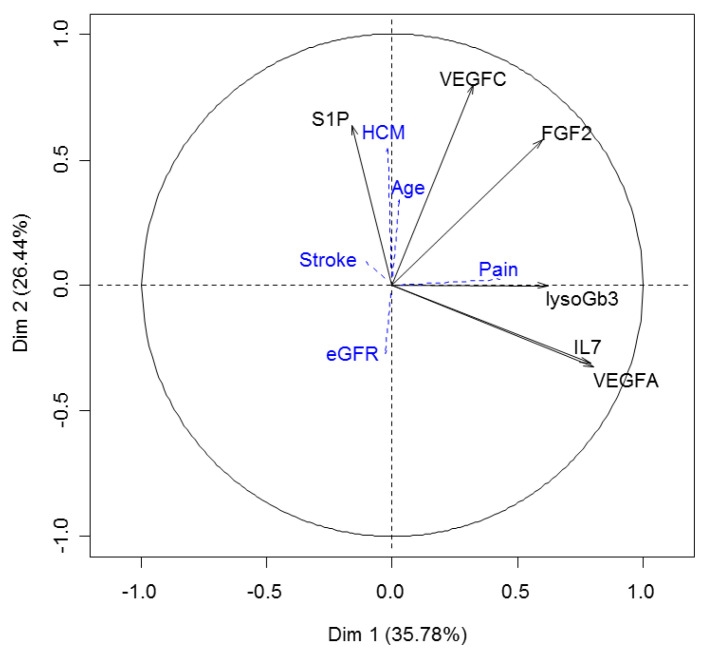
Variables factor map of principal component analysis including S1P, lysoGb3, FGF2, VEGF-A, VEGF-C, and IL-7 as active variables. Acral pain (Pain), hypertrophic cardiomyopathy (HCM), cerebral stroke, and estimated glomerular filtration rate (eGFR) are included as illustrative variables.

**Figure 3 jcm-11-01233-f003:**
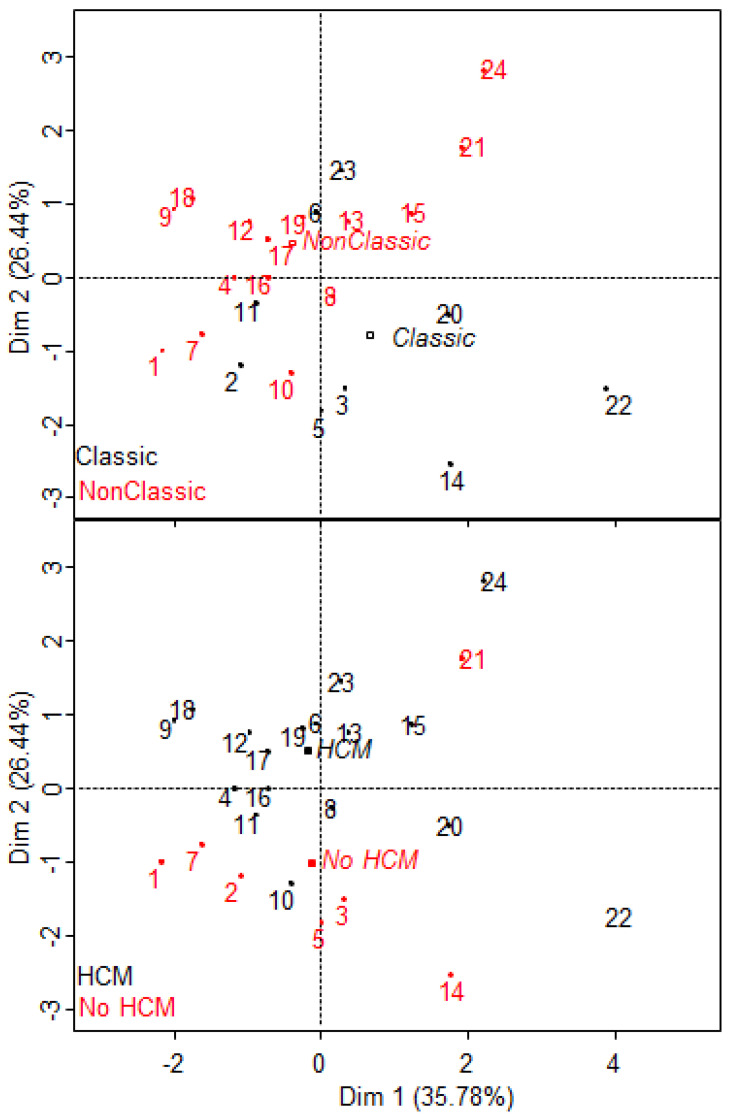
Individuals and qualitative factor map of principal component analysis including S1P, lysoGb3, FGF2, VEGF-A, VEGF-C, and IL-7 as active variables. Phenotype (Classic or Non-classic) (upper) and hypertrophic cardiomyopathy (HCM) (lower) are included as illustrative variables.

**Table 1 jcm-11-01233-t001:** Characteristics of patients (median [interquartile]; eGFR = estimated glomerular filtration rate according to the CKD-EPI equation; IST: interventricular septum thickness; MSSI Mainz Severity Score Index).

	Classic	Non-Classic	All
N	20	21	41
Age at diagnosis (years)	23.7 [16.8–35.8]	45.4 [32.2–55.3]	32.7 [20.0–49.2]
Age at sampling	37.6 [29.5–46.2]	48.1 [43.0–59.8]	45.3 [33.2–51.8]
Patients treated (n)	18	16	34
Cumulative treatment exposure (years)	11.4 [5.6–13.0]	4.4 [1.3–7.5]	6.6 [3.8–12.8]
eGFR in mL/min/1,73 m^2^	104.9 [61.8–118.2]	99.5 [65.9–114–4]	102.7 [61.8–118.2]
Kidney transplant (n)	4	0	4
Hypertrophic cardiomyopathy (n)	7	18	25
Arrhythmia (n)	7	12	19
Interventricular Septum Thickness (mm)	11 [9–15]	18 [14–22]	14 [11–22]
recent IST assessment available in n	12	12	24
Ischemic stroke	1	1	2
White matter lesions n/n with brain MRI	4/12	4/9	8/21
FFABRY Heart score	1 [0]	2 [1–3]	1 [1–3]
FFABRY Kidney score	1 [0–4]	0 [0–2]	1 [0–2]
FFABRY Neurological score	1 [0.1]	0 [0.1]	1 [0.1]
FFABRY Total score	3 [1–6]	4 [2–6]	3.5 [1–6]
MSSI cardiovascular	2 [0–6.25]	9 [2–13]	3 [0–12]
MSSI renal	2 [0–8]	0 [0–8]	0 [0–8]
MSSI neurological	6 [2–9.5]	2 [0–5]	5 [1–8]
MSSI general	4.5 [3.5–7.5]	2 [1–4]	4 [2–6]
MSSI total	19.5 [13.5–29.5]	20 [12–24]	20 [12–26]
LysoGb3 (ng/mL)	18.9 [10.6–48.8]	7.1 [2.6–22.1]	13.8 [6.5–31.8]
Treated patients (n)	18.7 [10.5–43.0] (17)	7.1 [2.6–22.1] (15)	
Untreated patients (n)	109 (1)	5.6 [2.6–24.6] (4)	

**Table 2 jcm-11-01233-t002:** Logistic regression analyses (S1P: Sphingosine-1-phosphate).

Risk of Hypertrophic Cardiomyopathy	Odds Ratio	Lower 95%CI	Upper 95%CI
multivariate analysis (age, lysoGb3, S1P, Phenotype)			
(Intercept)	0.00000038	1.93 × 10^−12^	0.075
age	1.33000000	1.05	1.690
lysoGb3	1.05000000	0.979	1.130
S1P	1.01000000	0.986	1.030
Phenotype (Non-Classic)	16.20000000	0.409	643.000
univariate analysis			
age	1.250000	1.090000000	1.4300
lysoGb3	0.997	0.971	1.02
S1P	1.010	0.9990	1.02
Phenotype (Non-Classic)	10.300	2.21	47.80
Risk of classic phenotype	odds ratio	Lower 95%CI	Upper 95%CI
**multivariate analysis (age, lysoGb3, S1P and treatment exposure)**			
(Intercept)	111.000	0.401	30,700.000
age	0.961	0.890	1.040
lysoGb3	1.040	0.992	1.080
S1P	0.974	0.951	0.998
cumulative treatment exposure	1.160	0.980	1.380
**multivariate analysis (age, lysoGb3, S1P and treatment exposure) with stepwise selection based on *p*-value**			
(Intercept)	96.50	1.040	8940.000
lysoGb3	1.05	1.010	1.100
S1P	0.97	0.946	0.994

## Data Availability

Data are available upon request to W. Mauhin.

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
