# Peer review of "Sphingosine-1-Phosphate Levels Are Higher in Male Patients with Non-Classic Fabry Disease"

_jcm, 2022, doi:10.3390/jcm11051233_

Round 1

Reviewer 1 Report

Sphingosine-1-phosphate is higher in male patients with non-classic Fabry disease.

The authors present data indicating that S1P serum levels are significantly higher in non-classic compared to classic Fabry disease patients suggesting involvement of distinct pathophysiological pathways between phenotypes.

I would like the authors respond to the following comments:

  1. In the classic group, there is a 14-year gap between the age at diagnosis and age at sampling, together with a cumulative treatment exposure of 11,4 compared to 4,4 years in the non-classic group. Could this influence the difference in S1P values?
  2. There must be someone with eGFR <90, because the range is 65,9-114,4. Then, I think that the MSSI renal must not be zero.
  3. Please, give the values of Lyso-Gb3 for the group of classic and non-classic patients. In addition, it would be interesting to know if the patients are treated or not, because lyso-Gb3 values vary with ERT or migalastat.
  4. Lyso-Gb3 was performed in the same sample than S1P after many years of treatment? We know that lyso-Gb3 varies with the time to treatment exposure, but we don't know what happens with S1P.
  5. It is a little bit puzzling the different number of samples that were used for the different determinations of S1P, (41) lyso-Gb3 (36) and the proteomic signature (24). This work was done on 41 male patients, but we don't know the correspondence with the 36 lyso-Gb3 samples, nor with the 24 samples for proteomic analysis, coming from a previous study in which there were male and female Fabry patients.
  6. The authors do not present controls of S1P values for their protocol, which uses a different methodology than referred previous publications.
  7. If the patient #21 is a classic Fabry type, why he appears in red (Non classic group)?
  8. S1P levels were not correlated with age, and the cohort of this study included all the available males with different disease stages and treatments, thus it seems daring the pretension of identify the pre-symptomatic cardiopathy.
  9. How do you explain that S1P alone did not appear sufficient to discriminate patients with cardiomyopathy, although S1P serum level is significantly higher in non-classic (where most patients have HCM) compared to classic Fabry disease patients?

Author Response

Answers to Reviewer 1

We thank the reviewer for his careful study of our work. We have taken into account all the questions raised and modified our manuscript. We gratefully thank the reviewer thinking that these modifications clearly improve our manuscript.

Question 1: In the classic group, there is a 14-year gap between the age at diagnosis and age at sampling, together with a cumulative treatment exposure of 11,4 compared to 4,4 years in the non-classic group. Could this influence the difference in S1P values?

Indeed, this is a very important point. Historically, our patients diagnosed with Fabry disease have a classic phenotype. The description of non-classic Fabry disease is quite recent. Therefore, our oldest patients are of classic phenotype and are treated. We have addressed this question in our statistical analysis. It appears that neither the age nor the treatment influence S1P levels.

Text Line 233-234: “S1P concentration was not influenced by the delay from sampling blood to thawing serum (Spearman p = 0.9), by the patient’s treatment (p = 0.8)”

To clarify, we modify the previous sentence: “S1P concentration was not influenced by the delay from sampling blood to thawing serum (Spearman p = 0.9), by the cumulative exposure to treatment (p = 0.8)”

Text Line 239: “S1P levels did not correlate with age (r=0.2; p=0.2)” and figure 1a

Question 2: There must be someone with eGFR <90, because the range is 65,9-114,4. Then, I think that the MSSI renal must not be zero.

If the reviewer’s comment deals with the table 1, the range is an “interquartile range” classically used to express the statistical dispersion of data along with the median. Also, the MSSI score is inversely correlated with the eGFR (MSSI 0 = no renal involvement). MSSI 0 corresponds to eGFR > 90 ml/min without proteinuria. We verified and confirm this range.

Question 3: Please, give the values of Lyso-Gb3 for the group of classic and non-classic patients. In addition, it would be interesting to know if the patients are treated or not, because lyso-Gb3 values vary with ERT or migalastat.

Indeed, the reviewer is right, lysoGb3 levels are influenced by the phenotype and the treatment. Also, we know that the improvement of lysoGb3 is not continuous and that lysoGb3 levels rapidly reach a plateau. This issue has been addressed in the supplementary table where lysoGb3 levels, S1P levels and treatment are mentioned. To clarify this key point, we modified the table 1 to include lysoGb3 levels depending on phenotype and treatment.

LysoGb3 (ng/ml)

18.9 [10.6-48.8]

7.1 [2.6-22.1]

13.8 [6.5-31.8]

Treated patients (n)

18.7 [10.5-43.0] (17)

7.1 [2.6-22.1] (15)

Untreated patients (n)

109 (1)

5.6  [2.6-24.6] (4)

Question 4: Lyso-Gb3 was performed in the same sample than S1P after many years of treatment? We know that lyso-Gb3 varies with the time to treatment exposure, but we don't know what happens with S1P.

Yes, lysoGb3 and S1P have been determined on the same sample. Lyso-Gb3 level reaches a plateau and does not improve whatever the time to treatment exposure in the absence of anti-drug antibodies.

To address this question regarding S1P, we performed statistical analyses and observed that:

  • S1P levels was not influenced […] by the patient’s treatment (Mann Whitney 200.3 ng/ml vs 192.8; p = 0.8; line 90)
  • S1P levels were not correlated with the time to treatment exposure (Spearman r = -0.09, p=0.6) not shown
  • S1P levels were not correlated with the time to treatment exposure in classic patients (r=0.08; p=0.7) not shown
  • S1P levels were not correlated with the time to treatment exposure in non-classic patients (r=-0.45; p=0.5) not shown
  • S1P levels were not correlated with the presence of anti-agalsidase antibodies (p = 0.6; line 91)

After this univariate approach, we performed multivariate analyses as detailed in the manuscript.

We have tried to clarify the text in the manuscript.

“Median serum S1P concentration was 193.2 ng/ml [IQ: 168.6 – 219.4]. S1P concentration was not influenced by the delay from sampling blood to thawing serum (Spearman p = 0.9), by the patient’s treatment (Mann Whitney: median S1P 200.3 ng/ml in untreated patients vs 192.8 ng/ml in treated patients; p = 0.8) or by the presence of anti-agalsidase antibodies (p = 0.6). S1P levels were not correlated with the time to treatment exposure (Spearman p = 0.6).”

Lines: 233-238

Question 5: It is a little bit puzzling the different number of samples that were used for the different determinations of S1P, (41) lyso-Gb3 (36) and the proteomic signature (24). This work was done on 41 male patients, but we don't know the correspondence with the 36 lyso-Gb3 samples, nor with the 24 samples for proteomic analysis, coming from a previous study in which there were male and female Fabry patients.

We agree with the reviewer that this can be confusing and we have mentioned all the correspondences in the supplementary material. The present work only deals with males to limit the heterogeneity.

Question 6: The authors do not present controls of S1P values for their protocol, which uses a different methodology than referred previous publications.

This is true. This is a limitation of our study. Nevertheless, the main goal of our work is not to assess S1P levels compared to normal controls but to assess the difference between classic and non-classic Fabry patients. Mirzaian et al. reported in 2016 that S1P levels of non Fabry controls was not different from classic Fabry patients.

This has been mentioned in the text Another study from Mirzaian et al., conducted with classic Fabry patients did not report obvious increase in plasma S1P compared to non Fabry individuals”

  1. Mirzaian, M.; Wisse, P.; Ferraz, M.J.; Marques, A.R.A.; Gabriel, T.L.; van Roomen, C.P.A.A.; Ottenhoff, R.; van Eijk, M.; Codée, J.D.C.; van der Marel, G.A.; et al. Accurate Quantification of Sphingosine-1-Phosphate in Normal and Fabry Disease Plasma, Cells and Tissues by LC-MS/MS with (13)C-Encoded Natural S1P as Internal Standard. Clin. Chim. Acta. 2016, 459, 36–44, doi:10.1016/j.cca.2016.05.017.

Question 7: If the patient #21 is a classic Fabry type, why he appears in red (Non-classic group)?

If the reviewer mentions the numbers listed in the first column of the supplementary data, there are different from the ones in the figure 3. The identification numbers in the figure 3 are made to compare phenotype and cardiopathy.

We had the following note to clarify this point:

“* Patient identification numbers are different from the figure 3”

Question 8: S1P levels were not correlated with age, and the cohort of this study included all the available males with different disease stages and treatments, thus it seems daring the pretension of identify the pre-symptomatic cardiopathy.

We totally agree with the reviewer and we may not have been clear enough: “S1P alone did not appear sufficient to discriminate patients with cardiomyopathy.” Line 210. We think that S1P is associated with the non-classic phenotype and thus may play a role in the pathophysiology of non-classic Fabry disease. We think that the management of Fabry disease should be personalized: non-classic Fabry disease patients usually have an almost isolated late-onset hypertrophic cardiopathy whereas classic Fabry disease patients develop, in addition to the early cardiopathy, renal disease, higher risk for stroke, pain… S1P along with other markers (lysoGb3, VEGF, FGF2, IL-7) may be useful to classify patients and better stratify the risk, before the development of any symptom, allowing a personalized management.

To clarify, we modified the text:

“S1P alone did not appear sufficient to discriminate patients with cardiomyopathy. Nevertheless, we observed that serum S1P associated to lysoGb3, VEGF-C, VEGF-A, IL-7 and FGF2 differentiate specific clusters of patients according to their phenotype, notably the presence of HCM. Further studies in different cohorts of patients are needed to validate these results and to determine if the 6 molecules may help in stratifying the risk to develop the different symptoms of Fabry disease, which could improve the management of presymptomatic Fabry patients.”

Lines: 466-473

Question 9: How do you explain that S1P alone did not appear sufficient to discriminate patients with cardiomyopathy, although S1P serum level is significantly higher in non-classic (where most patients have HCM) compared to classic Fabry disease patients?

This is a very important point and this is the very message of our work: we think that S1P plays a role in the pathophysiology of non-classic Fabry disease.

First, I would do a parallel with plasma lysoGb3. LysoGb3 is already significantly higher in classic Fabry patients before the development of the renal disease. We think that this is the same for S1P in non-classic Fabry patients.

To date, the difference in residual enzymatic activity is believed to be the only factor that explains the different phenotypes of Fabry disease. Considering this, it is intriguing to observe that there is no renal involvement in the non-classic phenotype whereas the kidney appears to be the most sensitive organ to the lack of alpha-galactosidase A. In others words, if the problem would only be a question of enzyme activity, non-classic patients should develop a renal disease… We believe that other pathophysiological pathways play some role in the development of the Fabry cardiopathy. We believe that S1P is a key-player of these pathways.

Reviewer 2 Report

The paper

Sphingosine-1-phosphate is higher in male patients with non-classic Fabry disease.

should be accepted because it opens up an interesting perspective for personalized treatment of Fabry disease and a debate among clinicians and researchers.

The weakness of the paper is represented by the scarcity of the cases that were analyzed.  The authors are aware of this and they state: “The major limitation of this study is the heterogeneity of patients in terms of age and the prevalence of cardiac disease between phenotype groups. However, this limitation is inherent in studies in rare diseases.”

Fabry disease is a rare disease with added difficulty. More than 1000 genotypes are known and there is no prevailing genotype. Admittedly papers based on data of patients suffer from great heterogeneity.

However, some changes and additions are necessary. The introduction is too succinct, and it is insufficient for most readers. For example, it should be mentioned that Fabry disease is characterized by a broad phenotypic and genotypic spectrum, that three drugs are approved for the treatment of patients, two recombinant enzymes, and a pharmacological chaperone. Biomarkers are a big issue. Some more papers should be mentioned regarding the difficulty of finding a marker to be used for diagnosis, but also for evaluating the effects of therapy.

I suggest these works, but others should be  added too by the authors

Citro, Valentina, et al. "The large phenotypic spectrum of Fabry disease requires graduated diagnosis and personalized therapy: A meta-analysis can help to differentiate missense mutations." International journal of molecular sciences 17.12 (2016):

Rubino, M. "Diagnosis and Management of Cardiovascular Involvement in Fabry Disease." Heart Failure Clinics 18.1 (2022): 39-49. Web.

McCafferty, Emma H., and Lesley J. Scott. "Migalastat: a review in Fabry disease." Drugs 79.5 (2019): 543-554.

Germain, Dominique P., et al. "The effect of enzyme replacement therapy on clinical outcomes in male patients with Fabry disease: A systematic literature review by a European panel of experts." Molecular genetics and metabolism reports 19 (2019): 100454.

Simonetta, Irene, et al. "Biomarkers in Anderson–Fabry Disease." International Journal of Molecular Sciences 21.21 (2020): 8080.

I have noticed that patients with the same genotype can have different phenotypes. Even if the number of cases is insufficient to have statistically significant results, it could be interesting to calculate the average value of Sphingosine-1-phosphate concentration for patients Arg112Cys, Arg301Gln, Gln283 *, Trp162Cys, Trp162Cys Classic and nonClassic.

In the supplementary table that summarizes the data for each patient, I did not find the AGAL activity. It would be interesting to correlate Sphingosine-1-phosphate concentration with the residual enzymatic activity measured in patients for the two phenotypes. Alternatively, the residual activity values measured in vitro for the various genotypes obtained from Fabry_cep

Cammisa, Marco, et al. "Fabry_CEP: a tool to identify Fabry mutations responsive to pharmacological chaperones." Orphanet journal of rare diseases 8.1 (2013): 1-3.

can be used. In this case, the residual activity could be set equal to 0 in cases of nonsense or frameshift mutations.

I was struck by the presence of patients with nonsense mutations, frameshift deletions with nonClassic phenotype. The authors can comment on this?

Author Response

Answers to Reviewer 2

Sphingosine-1-phosphate is higher in male patients with non-classic Fabry disease.

should be accepted because it opens up an interesting perspective for personalized treatment of Fabry disease and a debate among clinicians and researchers.

We thank the reviewer for the kind comment.

The weakness of the paper is represented by the scarcity of the cases that were analyzed.  The authors are aware of this and they state: “The major limitation of this study is the heterogeneity of patients in terms of age and the prevalence of cardiac disease between phenotype groups. However, this limitation is inherent in studies in rare diseases.”

Fabry disease is a rare disease with added difficulty. More than 1000 genotypes are known and there is no prevailing genotype. Admittedly papers based on data of patients suffer from great heterogeneity.

Indeed we are aware of the heterogeneity of our patients. We also performed multivariate analysis to limit the bias.

However, some changes and additions are necessary. The introduction is too succinct, and it is insufficient for most readers. For example, it should be mentioned that Fabry disease is characterized by a broad phenotypic and genotypic spectrum, that three drugs are approved for the treatment of patients, two recombinant enzymes, and a pharmacological chaperone. Biomarkers are a big issue. Some more papers should be mentioned regarding the difficulty of finding a marker to be used for diagnosis, but also for evaluating the effects of therapy.

We thank the reviewer for this suggestion. We have broadly changed the introduction, including numerous citations (in bold) as follows:

Lines: 57-107

“Fabry disease (FD, OMIM #301500) is an X-linked disorder characterized by defects in the alpha-galactosidase A enzyme activity that leads to an ubiquitous accumulation of glycosphingolipids, mainly of globotriaosylceramide (Gb3) and globotriaosylsphingosine (lysoGb3). Depending on the alpha-galactosidase A gene (GLA, Xq22.1 300,644) variant, two main phenotypes have been described [1]. The historical classic disease is associated with markedly reduced or absent enzyme activity and a wide spectrum of symptoms including acral neuropathic pain, progressive renal failure, white matter lesions, cerebral stroke and hypertrophic cardiomyopathy (HCM). The non-classic phenotype is observed in patients with low but detectable enzyme activity and characterized by an almost exclusive cardiomyopathy [1,2]. According to the last expert consensus document on Fabry disease edited by the European Society of Cardiology, Fabry disease could explain up to 1% of unexplained hypertrophy [3]. More than 1000 pathogenic variants of the GLA gene have been described[4]. Whereas deletions, frameshifts and nonsense mutations of the GLA gene have been associated with the classic phenotype, the genotype-phenotype correlation is more unobvious with for missense variants[1,5].

To date, two different therapeutic options have been validated in Fabry disease. Intravenous enzyme replacement therapy (ERT) with agalsidase alfa (Replagal®, Shire-Takeda) or agalsidase beta (Fabrazyme®, Sanofi-Genzyme) has been available since 2000 for all patients with Fabry disease [6,7]. The benefits of ERT have now been accepted despite the disease seems to progress in some patients [8,9]. The development of anti-drug antibodies that occurs preferentially in the classic Fabry males but also inflammation and secondary fibrosis that would occur in all patients, have been suspected to explain the partial failure of enzyme replacement therapy[8–11]. Since 2016, another validated option consists in a chaperone molecule therapy named migalastat (Galafold®, Amicus Therapeutics) that can only be given to eligible patients defined by an amenable variant of the GLA gene, usually a missense mutation [12]. De facto, migalastat mainly concerns non-classic Fabry patients. Also, the definition of amenable variants – that is the ones associated with an increase of enzymatic alpha-galactosidase A activity in presence of migalastat – has been controversial[13,14].

Plasma globotriaosylsphingosine, also known as lysoGb3, is the deacetylated derivative from Gb3 [15]. Plasma lysoGb3 has been observed increased in Fabry patients, being higher in males than in females and higher in classic than in non-classic patients [16]. Although the level of plasma lysoGb3 can predict the clinical phenotype, it does not reflect the burden of the disease[16,17]. Hence, no ideal biomarker exists to manage Fabry disease [16,18]. A recent proteomics approach revealed that some angiogenesis proteins (fibroblast growth factor 2 (FGF2), vascular endothelial growth factor A (VEGFA), vascular endothelial growth factor C (VEGFC)) and one cytokine interleukin 7 (IL-7) were significantly higher in Fabry patients, independently from the lysoGb3 plasma levels [19]. Also, IL-7 was correlated with the residual enzyme activity in non-classic patients which may reflect a specific pathophysiology in the non-classic phenotype [19]. Plasma sphingosine-1-phosphate (S1P) has been controversially associated with Fabry cardiomyopathy [20,21]. S1P is implicated in cardiac hypertrophy, cardiac fibrosis and immune regulation [22]. Treatments targeting the S1P signaling pathway are currently used to control multiple sclerosis (FTY720, fingolimod) [23,24] and have shown promising results in hypertension (PF543) [25]. Whereas the existence of a pro-inflammatory condition in Fabry disease has now been accepted [11,26], we aimed to determine whether S1P levels could be of interest in the management of Fabry disease, notably for the cardiovascular involvement. We measured S1P in serum from male patients of the French multicenter cohort FFABRY and assessed statistical correlations with clinical data.”

  1. Brady, R.O.; Schiffmann, R. Clinical Features of and Recent Advances in Therapy for Fabry Disease. JAMA 2000, 284, 2771–2775, doi:10.1001/jama.284.21.2771.
  2. Rubino, M.; Monda, E.; Lioncino, M.; Caiazza, M.; Palmiero, G.; Dongiglio, F.; Fusco, A.; Cirillo, A.; Cesaro, A.; Capodicasa, L.; et al. Diagnosis and Management of Cardiovascular Involvement in Fabry Disease. Heart Fail. Clin. 2022, 18, 39–49, doi:10.1016/j.hfc.2021.07.005.
  3. Linhart, A.; Germain, D.P.; Olivotto, I.; Akhtar, M.M.; Anastasakis, A.; Hughes, D.; Namdar, M.; Pieroni, M.; Hagège, A.; Cecchi, F.; et al. An Expert Consensus Document on the Management of Cardiovascular Manifestations of Fabry Disease. Eur. J. Heart Fail. 2020, 22, 1076–1096, doi:10.1002/ejhf.1960.
  4. Germain, D.P.; Levade, T.; Hachulla, E.; Knebelmann, B.; Lacombe, D.; Seguin, V.L.; Nguyen, K.; Noël, E.; Rabès, J.-P. Challenging the Traditional Approach for Interpreting Genetic Variants: Lessons from Fabry Disease. Clin. Genet. 2021, doi:10.1111/cge.14102.
  5. Pan, X.; Ouyang, Y.; Wang, Z.; Ren, H.; Shen, P.; Wang, W.; Xu, Y.; Ni, L.; Yu, X.; Chen, X.; et al. Genotype: A Crucial but Not Unique Factor Affecting the Clinical Phenotypes in Fabry Disease. PloS One 2016, 11, e0161330, doi:10.1371/journal.pone.0161330.
  6. Eng, C.M.; Guffon, N.; Wilcox, W.R.; Germain, D.P.; Lee, P.; Waldek, S.; Caplan, L.; Linthorst, G.E.; Desnick, R.J.; International Collaborative Fabry Disease Study Group Safety and Efficacy of Recombinant Human Alpha-Galactosidase A Replacement Therapy in Fabry’s Disease. N. Engl. J. Med. 2001, 345, 9–16, doi:10.1056/NEJM200107053450102.
  7. Schiffmann, R.; Kopp, J.B.; Austin, H.A.; Sabnis, S.; Moore, D.F.; Weibel, T.; Balow, J.E.; Brady, R.O. Enzyme Replacement Therapy in Fabry Disease: A Randomized Controlled Trial. JAMA 2001, 285, 2743–2749, doi:10.1001/jama.285.21.2743.
  8. Lenders, M.; Brand, E. Mechanisms of Neutralizing Anti-Drug Antibody Formation and Clinical Relevance on Therapeutic Efficacy of Enzyme Replacement Therapies in Fabry Disease. Drugs 2021, 81, 1969–1981, doi:10.1007/s40265-021-01621-y.
  9. Germain, D.P.; Elliott, P.M.; Falissard, B.; Fomin, V.V.; Hilz, M.J.; Jovanovic, A.; Kantola, I.; Linhart, A.; Mignani, R.; Namdar, M.; et al. The Effect of Enzyme Replacement Therapy on Clinical Outcomes in Male Patients with Fabry Disease: A Systematic Literature Review by a European Panel of Experts. Mol. Genet. Metab. Rep. 2019, 19, 100454, doi:10.1016/j.ymgmr.2019.100454.
  10. Song, H.-Y.; Yang, Y.-P.; Chien, Y.; Lai, W.-Y.; Lin, Y.-Y.; Chou, S.-J.; Wang, M.-L.; Wang, C.-Y.; Leu, H.-B.; Yu, W.-C.; et al. Reversal of the Inflammatory Responses in Fabry Patient IPSC-Derived Cardiovascular Endothelial Cells by CRISPR/Cas9-Corrected Mutation. Int. J. Mol. Sci. 2021, 22, 2381, doi:10.3390/ijms22052381.
  11. Rozenfeld, P.; Feriozzi, S. Contribution of Inflammatory Pathways to Fabry Disease Pathogenesis. Mol. Genet. Metab. 2017, 122, 19–27, doi:10.1016/j.ymgme.2017.09.004.
  12. Germain, D.P.; Hughes, D.A.; Nicholls, K.; Bichet, D.G.; Giugliani, R.; Wilcox, W.R.; Feliciani, C.; Shankar, S.P.; Ezgu, F.; Amartino, H.; et al. Treatment of Fabry’s Disease with the Pharmacologic Chaperone Migalastat. N. Engl. J. Med. 2016, 375, 545–555, doi:10.1056/NEJMoa1510198.
  13. Cammisa, M.; Correra, A.; Andreotti, G.; Cubellis, M.V. Fabry_CEP: A Tool to Identify Fabry Mutations Responsive to Pharmacological Chaperones. Orphanet J. Rare Dis. 2013, 8, 111, doi:10.1186/1750-1172-8-111.
  14. Lenders, M.; Stappers, F.; Niemietz, C.; Schmitz, B.; Boutin, M.; Ballmaier, P.J.; Zibert, A.; Schmidt, H.; Brand, S.-M.; Auray-Blais, C.; et al. Mutation-Specific Fabry Disease Patient-Derived Cell Model to Evaluate the Amenability to Chaperone Therapy. J. Med. Genet. 2019, 56, 548–556, doi:10.1136/jmedgenet-2019-106005.
  15. Aerts, J.M.; Groener, J.E.; Kuiper, S.; Donker-Koopman, W.E.; Strijland, A.; Ottenhoff, R.; van Roomen, C.; Mirzaian, M.; Wijburg, F.A.; Linthorst, G.E.; et al. Elevated Globotriaosylsphingosine Is a Hallmark of Fabry Disease. Proc. Natl. Acad. Sci. U. S. A. 2008, 105, 2812–2817, doi:10.1073/pnas.0712309105.
  16. Bichet, D.G.; Aerts, J.M.; Auray-Blais, C.; Maruyama, H.; Mehta, A.B.; Skuban, N.; Krusinska, E.; Schiffmann, R. Assessment of Plasma Lyso-Gb3 for Clinical Monitoring of Treatment Response in Migalastat-Treated Patients with Fabry Disease. Genet. Med. Off. J. Am. Coll. Med. Genet. 2021, 23, 192–201, doi:10.1038/s41436-020-00968-z.
  17. Nowak, A.; Beuschlein, F.; Sivasubramaniam, V.; Kasper, D.; Warnock, D.G. Lyso-Gb3 Associates with Adverse Long-Term Outcome in Patients with Fabry Disease. J. Med. Genet. 2021, jmedgenet-2020-107338, doi:10.1136/jmedgenet-2020-107338.
  18. Simonetta, I.; Tuttolomondo, A.; Daidone, M.; Pinto, A. Biomarkers in Anderson-Fabry Disease. Int. J. Mol. Sci. 2020, 21, E8080, doi:10.3390/ijms21218080.
  19. Tebani, A.; Mauhin, W.; Abily-Donval, L.; Lesueur, C.; Berger, M.G.; Nadjar, Y.; Berger, J.; Benveniste, O.; Lamari, F.; Laforêt, P.; et al. A Proteomics-Based Analysis Reveals Predictive Biological Patterns in Fabry Disease. J. Clin. Med. 2020, 9, E1325, doi:10.3390/jcm9051325.
  20. Brakch, N.; Dormond, O.; Bekri, S.; Golshayan, D.; Correvon, M.; Mazzolai, L.; Steinmann, B.; Barbey, F. Evidence for a Role of Sphingosine-1 Phosphate in Cardiovascular Remodelling in Fabry Disease. Eur. Heart J. 2010, 31, 67–76, doi:10.1093/eurheartj/ehp387.
  21. Mirzaian, M.; Wisse, P.; Ferraz, M.J.; Marques, A.R.A.; Gabriel, T.L.; van Roomen, C.P.A.A.; Ottenhoff, R.; van Eijk, M.; Codée, J.D.C.; van der Marel, G.A.; et al. Accurate Quantification of Sphingosine-1-Phosphate in Normal and Fabry Disease Plasma, Cells and Tissues by LC-MS/MS with (13)C-Encoded Natural S1P as Internal Standard. Clin. Chim. Acta Int. J. Clin. Chem. 2016, 459, 36–44, doi:10.1016/j.cca.2016.05.017.
  22. Jozefczuk, E.; Guzik, T.J.; Siedlinski, M. Significance of Sphingosine-1-Phosphate in Cardiovascular Physiology and Pathology. Pharmacol. Res. 2020, 156, 104793, doi:10.1016/j.phrs.2020.104793.
  23. Baer, A.; Colon-Moran, W.; Bhattarai, N. Characterization of the Effects of Immunomodulatory Drug Fingolimod (FTY720) on Human T Cell Receptor Signaling Pathways. Sci. Rep. 2018, 8, 10910, doi:10.1038/s41598-018-29355-0.
  24. Ward, M.D.; Jones, D.E.; Goldman, M.D. Overview and Safety of Fingolimod Hydrochloride Use in Patients with Multiple Sclerosis. Expert Opin. Drug Saf. 2014, 13, 989–998, doi:10.1517/14740338.2014.920820.
  25. Józefczuk, E.; Nosalski, R.; Saju, B.; Crespo, E.; Szczepaniak, P.; Guzik, T.J.; Siedlinski, M. Cardiovascular Effects of Pharmacological Targeting of Sphingosine Kinase 1. Hypertens. Dallas Tex 1979 2020, 75, 383–392, doi:10.1161/HYPERTENSIONAHA.119.13450.
  26. Mauhin, W.; Lidove, O.; Masat, E.; Mingozzi, F.; Mariampillai, K.; Ziza, J.-M.; Benveniste, O. Innate and Adaptive Immune Response in Fabry Disease. JIMD Rep. 2015, 22, 1–10, doi:10.1007/8904_2014_371.

I have noticed that patients with the same genotype can have different phenotypes. Even if the number of cases is insufficient to have statistically significant results, it could be interesting to calculate the average value of Sphingosine-1-phosphate concentration for patients Arg112Cys, Arg301Gln, Gln283 *, Trp162Cys, Trp162Cys Classic and nonClassic.

We thank the reviewer for this interesting suggestion. The small number of patients prevents any statistical analysis but we observed that S1P was higher in non-classic compared to classic patients in pairs sharing the same genetic variant: (p.Phe337Ser: 229.5 vs 200.98 ng/ml respectively in non-classic and classic patients; p.Trp162Cys: 230.2 vs 92.3 ng/ml respectively in non-classic and classic patients; p.Arg112Cys: 219.44 vs 190.7 ng/ml respectively in non-classic and classic patients; c.802-3_802-2del/ p? 268.1 vs 125.5 respectively in non-classic and classic patients; p.Arg301Gln: 219.2 vs 210.2 respectively in non-classic and classic patients). The only mismatch concerns the p.Gln283* genotype with a renal transplant classic male had a S1P level of 234.5 ng/ml compared to the189.6 and 199.1 ng/ml observed in the 2 non-classic associated patients. We then observed that among all classic patients, kidney transplant patients had a trend to higher S1P levels compared to non-transplant patients (median 227.9 vs 163.6 ng/ml, p=0.1).

We had these data in our results and think that it improves our manuscript, as follows:

“In accordance with the possible genotype-phenotype heterogeneity, some patients – 5 pairs and 1 trio - shared the same genotype but a different phenotype. Although the small number of patients prevents any statistical conclusion, we observed that S1P was higher in non-classic compared to classic patients in pairs sharing the same genetic variant: (p.Phe337Ser: 229.5 vs 200.98 ng/ml respectively in non-classic and classic patients; p.Trp162Cys: 230.2 vs 92.3 ng/ml; p.Arg112Cys: 219.44 vs 190.7 ng/ml; c.802-3_802-2del/ p? 268.1 vs 125.5; p.Arg301Gln: 219.2 vs 210.2 ng/ml). The only mismatch concerns the p.Gln283* genotype: a renal transplant classic male had a S1P level of 234.5 ng/ml compared to 189.6 and 199.1 ng/ml observed in the 2 non-classic paired patients. We then observed that among all classic patients, kidney transplant patients had a trend to higher S1P levels compared to non-transplant patients (median 227.9 vs 163.6 ng/ml, p=0.1).” 

Lines: 360-371

In the supplementary table that summarizes the data for each patient, I did not find the AGAL activity. It would be interesting to correlate Sphingosine-1-phosphate concentration with the residual enzymatic activity measured in patients for the two phenotypes. Alternatively, the residual activity values measured in vitro for the various genotypes obtained from Fabry_cep

Cammisa, Marco, et al. "Fabry_CEP: a tool to identify Fabry mutations responsive to pharmacological chaperones." Orphanet journal of rare diseases 8.1 (2013): 1-3.

can be used. In this case, the residual activity could be set equal to 0 in cases of nonsense or frameshift mutations.

I was struck by the presence of patients with nonsense mutations, frameshift deletions with nonClassic phenotype. The authors can comment on this?

We agree with the reviewer and added this limitation in the discussion (unfortunately, the Fabry_CEP link seems not to work/ or was unavailable to us. We could not use it).

Lines: 457-465

“The phenotype classification was performed using the FFABRY algorithm that determines the phenotype on the presence of cornea verticillata and history of acroparesthesia in males. This classification takes in account the genotype-phenotype heterogeneity. The value of the residual enzymatic activity of alpha-galactosidase and plasma lysoGb3 at baseline, before any treatment could have allow a better classification. Unfortunately, many of the patients were treated for years, before the availability of plasma lysoGb3 dosage. Also, initial determination of alpha-galactosidase enzymatic activity had been defined in different places, using different techniques.”

Round 2

Reviewer 1 Report

Thank you for your comprehensive response. Still some points to clarify:

Question 2: There must be someone with eGFR <90, because the range is 65,9-114,4. Then, I think that the MSSI renal must not be zero.

If the reviewer’s comment deals with the table 1, the range is an “interquartile range” classically used to express the statistical dispersion of data along with the median. Also, the MSSI score is inversely correlated with the eGFR (MSSI 0 = no renal involvement). MSSI 0 corresponds to eGFR > 90 ml/min without proteinuria. We verified and confirmed this range.

What I see in your supplementary data is that there are at least 8 non-classic patients with eGFR lower than 90 ml/min/1.73 m2.

This also concerns with your final comment: “Considering this, it is intriguing to observe that there is no renal involvement in the non-classic phenotype whereas the kidney appears to be the most sensitive organ to the lack of alpha-galactosidase A”

Thus, apparently many non-classic FD patients may have a predominantly cardiac phenotype but renal involvement may appear although much late than in classic forms.

This has been mentioned in the text “Another study from Mirzaian et al., conducted with classic Fabry patients did not report obvious increase in plasma S1P compared to non Fabry individuals”

  1. Mirzaian, M.; Wisse, P.; Ferraz, M.J.; Marques, A.R.A.; Gabriel, T.L.; van Roomen, C.P.A.A.; Ottenhoff, R.; van Eijk, M.; Codée, J.D.C.; van der Marel, G.A.; et al. Accurate Quantification of Sphingosine-1-Phosphate in Normal and Fabry Disease Plasma, Cells and Tissues by LC-MS/MS with (13)C-Encoded Natural S1P as Internal Standard. Clin. Chim. Acta. 2016, 459, 36–44, doi:10.1016/j.cca.2016.05.017.

I think you have to admit and acknowledge that you can’t take for granted the previous results of S1P determined by LC-MS/MS compared to the present study performed with an ELISA methodology.

Author Response

Answers to reviewer 1

We thank again the reviewer 1 for his careful study of our work. We took into account all the comments and believe that our manuscript has improved again thanks to them.

Question 2: There must be someone with eGFR <90, because the range is 65,9-114,4. Then, I think that the MSSI renal must not be zero.

If the reviewer’s comment deals with the table 1, the range is an “interquartile range” classically used to express the statistical dispersion of data along with the median. Also, the MSSI score is inversely correlated with the eGFR (MSSI 0 = no renal involvement). MSSI 0 corresponds to eGFR > 90 ml/min without proteinuria. We verified and confirmed this range.

What I see in your supplementary data is that there are at least 8 non-classic patients with eGFR lower than 90 ml/min/1.73 m2.

This also concerns with your final comment: “Considering this, it is intriguing to observe that there is no renal involvement in the non-classic phenotype whereas the kidney appears to be the most sensitive organ to the lack of alpha-galactosidase A”

Thus, apparently many non-classic FD patients may have a predominantly cardiac phenotype but renal involvement may appear although much late than in classic forms.

This is true. Eight non-classic patients have an eGFR < 90 ml/min/1.73 m2. This explains the upper limit of the renal MSSI at 8 in the table 1.

In our clinical experience, in a tertiary center for Fabry disease, non-classic Fabry patients can effectively develop a kidney disease from the 5th decade. Nevertheless, non-classic patients who develop kidney disease do not have a typical Fabry nephropathy: most of them do not have proteinuria that we consider as a cornerstone of Fabry nephropathy.

If we admit that genotype-phenotype correlations remain sometimes controversial, elevated creatinine in non-classic Fabry patients is usually explained by the Fabry cardiopathy. The possible obstructive hypertrophic cardiopathy can cause cardio-renal syndrome. Also some of our patients have diabete, hypertension, obstructive uropathy or other evident cause of elevated creatinine. For example, the patient 41 with an eGFR of 27.5ml/min/1.73m² is a 74-year-old male suffering from a cardiac failure due to Fabry disease, severe hypertension and hepatitis B virus related cryoglobulinemic vasculitis.

We would be prone to perform a kidney biopsy in all non-classic Fabry patient with unexplained renal failure.

We had a comment in the supplementary table to clarify this point.

This has been mentioned in the text “Another study from Mirzaian et al., conducted with classic Fabry patients did not report obvious increase in plasma S1P compared to non Fabry individuals”

  1. Mirzaian, M.; Wisse, P.; Ferraz, M.J.; Marques, A.R.A.; Gabriel, T.L.; van Roomen, C.P.A.A.; Ottenhoff, R.; van Eijk, M.; Codée, J.D.C.; van der Marel, G.A.; et al. Accurate Quantification of Sphingosine-1-Phosphate in Normal and Fabry Disease Plasma, Cells and Tissues by LC-MS/MS with (13)C-Encoded Natural S1P as Internal Standard. Clin. Chim. Acta. 2016, 459, 36–44, doi:10.1016/j.cca.2016.05.017.

I think you have to admit and acknowledge that you can’t take for granted the previous results of S1P determined by LC-MS/MS compared to the present study performed with an ELISA methodology.

We agree with the reviewer. Both studies from Brakch and Mirzaian have been conducted with mass spectrometry, which is totally different from ELISA. We have added the following comments is this sense:

Lines 408 -409 “Both these studies used mass spectrometry to assess S1P levels. In our protocol using ELISA, the conditions for sampling, handling and storing serum were identical for all patients and we did not observe any influence of the time from sampling to thawing samples”

Lines 470-473 “Further studies in different cohorts of patients are needed to validate these results, notably using mass spectrometry, in order to determine if the 6 molecules may help in stratifying the risk to develop the different symptoms of Fabry disease, which could improve the management of presymptomatic Fabry patients.”

Reviewer 2 Report

Thanks for your response.

Author Response

We thank the reviewer for his careful study of our work. 

This manuscript is a resubmission of an earlier submission. The following is a list of the peer review reports and author responses from that submission.

Round 1

Reviewer 1 Report

In this study, Mauhin et al have compared serum levels of sphingosine-1-phosphate (S1P) in classic and non-classic phenotypes of Fabry patients and correlated these levels with various aspects of the disease. They report that S1P levels were significantly higher in non-classic Fabry patients suggesting a specific role of S1P in the pathogenesis of this form of the disease. Overall, the manuscript is clear. We have however found major limitations that reduce the strength of the manuscript:

  • Major differences exist between the classic and the non-classic groups. For instance only seven patients in the classic versus 18 patients in the non classic present an hypertrophic cardiomyopathy. Similarly, patients in the classic group were substantially younger. Hence, the severity score (MSSI total) was similar between both groups, contrasting with the fact that the classic form is generally more severe (multisystemic involvement) than the variant form.
  • Authors did not provide the Fabry mutations implicated in this study. This is important in order to understand why the phenotype score (MSSI total) is identical between both groups. This, in fact, probably reflects that patients in the classic group are younger. In addition, what do the authors mean by “S1P was not different between missense and non-sense GLA variant carriers”. As obviously non-sense GLA variant are mostly found in the classic group and missense in the non classic.
  • In figure 1c surprisingly only 12 patients are analyzed for each group presumably due to a lack of recent echocardiographic analysis for certain patients. Thus, this introduces a major bias and figure 1c is the major finding of the study. Similarly, lysoGB3 serum levels were not measured in every patients.
  • There is no control group. We therefore do not know whether the S1P values obtained in their cohort are abnormal or not.
  • It is well established that non classic male Fabry patients develop HCM about ten years later than classic patients. Accordingly, we do not fully understand how the authors made the hypothesis that S1P levels may be different between both groups and consequently the aim of the study is not clear. Consequently, we are missing the novelty of the study.

Author Response

Reviewer #1

In this study, Mauhin et al have compared serum levels of sphingosine-1-phosphate (S1P) in classic and non-classic phenotypes of Fabry patients and correlated these levels with various aspects of the disease. They report that S1P levels were significantly higher in non-classic Fabry patients suggesting a specific role of S1P in the pathogenesis of this form of the disease. Overall, the manuscript is clear. We have however found major limitations that reduce the strength of the manuscript:

  1. Major differences exist between the classic and the non-classic groups. For instance only seven patients in the classic versus 18 patients in the non classic present an hypertrophic cardiomyopathy. Similarly, patients in the classic group were substantially younger. Hence, the severity score (MSSI total) was similar between both groups, contrasting with the fact that the classic form is generally more severe (multisystemic involvement) than the variant form.

Response :

Indeed, we agree with the reviewer when he raised the points that major differences exist between the classic and the non-classic group and this reflects the major differences between these two phenotypes that are observed in the real life. Our cohort has already been described in the article from Mauhin et al, in PlosOne in 2020 (Cornea verticillata and acroparesthesia efficiently discriminate clusters of severity in Fabry disease). In this previous paper, we demonstrated that classic patients are diagnosed younger than non-classic patients due to the presence of acroparesthesia but also to the renal involvement (neurological MSSI (2 vs. 0) and renal MSSI (6 vs. 2) are therefore higher in classic vs. non–classic patients). Hence, our classic patients are younger (37.6 vs. 48.1 years old. We already published that the development of hypertrophic cardiomyopathy occurs earlier in classic patients. In our cohort, the median survival without hypertrophic cardiomyopathy was 46.3 years in classic vs. 57.6 years in non-classic patients. Being diagnosed younger, the cumulative exposure to treatment is also longer in the classic than in the non-classic patients (11.4 years vs. 4.4 years) with possible benefits in preventing or delaying the development of hypertrophic cardiomyopathy.

Also, in the previous paper, we discussed the pertinence of the total MSSI score that includes 26 variables, empirically weighted, among which hemorrhoids, facial appearance, and subjective fitness assessment. However, the MSSI does not include important items such as renal transplantation. That is the reason why we developed the FFABRY score. However, both FFABRY and MSSI scores are depending from the age and the phenotype of patients.

This difference in age was taken into account by including the age in the different logistic regression models. We agree with the reviewer if the comments are about the heterogeneity of patients. What is very interesting is that despite this heterogeneity in terms of severity into the different phenotypic groups, S1P level is not correlated with the age and so S1P level is not correlated to the severity. This suggests that S1P is associated with the phenotype by itself (and possibly implicated in the pathophysiology) rather than to the disease severity. However, this heterogeneity is inherent in rare diseases studies. This limitation was mentioned in line 221 “The major limitation of this study is the heterogeneity of patients in terms of age and the prevalence of cardiac disease between phenotype groups. However, this limitation is inherent in studies in rare diseases.”

  1. Authors did not provide the Fabry mutations implicated in this study. This is important in order to understand why the phenotype score (MSSI total) is identical between both groups. This, in fact, probably reflects that patients in the classic group are younger. In addition, what do the authors mean by “S1P was not different between missense and non-sense GLA variant carriers”. As obviously non-sense GLA variant are mostly found in the classic group and missense in the non classic.

Response:

We have provided a supplementary table with the different molecular variants and characteristics of patients to explain all the differences that are discussed in this point.

  1. In figure 1c surprisingly only 12 patients are analyzed for each group presumably due to a lack of recent echocardiographic analysis for certain patients. Thus, this introduces a major bias and figure 1c is the major finding of the study. Similarly, lysoGB3 serum levels were not measured in every patients.

Response:

This is true and has been mentioned several times throughout the manuscript. Only twenty-four patients had a recent evaluation of interventricular septum thickness (IST) by cardiac echography. Whereas the enzyme replacement therapy and the chaperone molecule can rapidly decrease the IST, it appeared important not to use old data. We could have restrained our study to the only 24 patients, nevertheless, the existence of a cardiac hypertrophy could be assessed with different tools such as cardiac MRI (as mentioned in Material and Methods). We first used Mann Whitney test to compare S1P levels between groups according to the existence of a hypertrophy. We then used a multivariate logistic regression model to explore the association of cardiac hypertrophy with S1P levels. We think that the most important data is the fact that the only significant variable associated with cardiac hypertrophy is the age and not S1P levels.

In 2010, Brakch et al (Eur. Heart J. 2010, 31, 67–76) reported in the European Heart Journal that S1P was correlated with cardiac hypertrophy using data from 17 patients, mixing males (of unknown phenotype) and females. The clinical stratification between classic and non-classic was not usual at this time. Today, using hypertrophic status of 41 male patients or IST measure of 24 male patients by echography, we can affirm that S1P is not correlated with the IST but with the clinical phenotype. We think that this is a novel observation.

In order to clarify this point, we have added in the discussion (line 223): “We also regret the lack of exhaustive recent echocardiographic data, concomitant to the S1P sampling, that would have strengthened the power of the analyses.”

Brakch, N.; Dormond, O.; Bekri, S.; Golshayan, D.; Correvon, M.; Mazzolai, L.; Steinmann, B.; Barbey, F. Evidence for a Role of Sphingosine-1 Phosphate in Cardiovascular Remodelling in Fabry Disease. Eur. Heart J. 2010, 31, 67–76, doi:10.1093/eurheartj/ehp387.

  1. There is no control group. We therefore do not know whether the S1P values obtained in their cohort are abnormal or not.

Response:

This is true. This is a limitation of our study. Nevertheless, the main goal of our work is not to assess S1P levels compared to normal controls but to assess the difference between classic and non-classic Fabry patients. Mirzaian et al. reported in 2016 that S1P levels of non Fabry controls was not different from classic Fabry patients.

This has been mentioned in the text (lines 170-174): Another study from Mirzaian et al., conducted with classic Fabry patients did not report obvious increase in plasma S1P compared to non Fabry individuals”

Mirzaian, M.; Wisse, P.; Ferraz, M.J.; Marques, A.R.A.; Gabriel, T.L.; van Roomen, C.P.A.A.; Ottenhoff, R.; van Eijk, M.; Codée, J.D.C.; van der Marel, G.A.; et al. Accurate Quantification of Sphingosine-1-Phosphate in Normal and Fabry Disease Plasma, Cells and Tissues by LC-MS/MS with (13)C-Encoded Natural S1P as Internal Standard. Clin. Chim. Acta. 2016, 459, 36–44, doi:10.1016/j.cca.2016.05.017.

  1. It is well established that non classic male Fabry patients develop HCM about ten years later than classic patients. Accordingly, we do not fully understand how the authors made the hypothesis that S1P levels may be different between both groups and consequently the aim of the study is not clear. Consequently, we are missing the novelty of the study.

Response:

Indeed, as mentioned earlier, HCM appears later in non-classic compared to classic Fabry patients. We already reported the characteristics of our cohort and this was our focus in this present work. Here, we wanted to highlight that S1P is of major interest in Fabry pathophysiology as S1P levels behave differently in classic and non-classic patients. S1P and other proteins recently discovered by targeted proteomics (VEGF-A and -C, FGF-2, IL-7), can offer new approaches to the comprehension of Fabry disease pathophysiology. Brakch et al. already reported that mice treated with S1P developed cardiac hypertrophy. Whereas therapeutics targeting S1P receptors already exist, it is of great interest to explore this pathway. We think that our work paves the way to these new approaches.

In order to clarify this important point we have added in the text (line 182-189): ” Aerts et al. demonstrated the pathogenic role of lysoGb3 on the smooth muscle cells of vessels with a proliferative effect. It is now admitted that plasma lysoGb3 is significantly lower in non-classic Fabry patients and even normal in some women. Brakch et al. demonstrated that S1P also has a proliferative effect and mice treated with S1P developed a Fabry-like hypertrophic cardiomyopathy. The results of these two experiments may explain the development of the Fabry cardiopathy in we consider that lysoGb3 and S1P play the major role respectively in the classic and non-classic phenotype.”

Aerts, J.M.; Groener, J.E.; Kuiper, S.; Donker-Koopman, W.E.; Strijland, A.; Ottenhoff, R.; van Roomen, C.; Mirzaian, M.; Wijburg, F.A.; Linthorst, G.E.; et al. Elevated Globotriaosylsphingosine Is a Hallmark of Fabry Disease. Proc. Natl. Acad. Sci. U. S. A. 2008, 105, 2812–2817, doi:10.1073/pnas.0712309105.

Reviewer 2 Report

Manuscript title: Sphingosine-1-phosphate is higher in the cardiac phenotype of Fabry disease (only cardiac phenotype or better “in male patients with non-classic type of Fabry disease”? If I’m not wrong, there are 3 non-classic patients without HCM. Are there’s S1P values lower than the rest of non-classic patients with HCM?)

S1P appears to be positively correlated with non-classic type of Fabry disease.

With respect to the patients’ characteristics, I would like to see the different genetic variants and the residual enzyme activity for missense mutations. I think that the classification in classic versus non-classic type will be more accurate using the residual enzyme activity.

In the present cohort of non-classic FD the majority have HCM (except 3).

Patients in the non-classic type of FD are older, and age is directly correlated with HCM.

Increase of IST may appear before HCM (what are the diagnostic criteria for HCM?)

Lyso-Gb3 is negatively correlated with the non-classic FD. Thus, apparently, S1P follows a different pathway as lyso-Gb3. Could you explain the relationship of S1P with alfa-Gal deficiency?

Could S1P elevation be an epiphenomenon secondary to lysosomal dysfunction or the inflammatory cascade? Is there any alteration of S1PRs that could produce an imbalance of S1P?

Author Response

  1. Manuscript title: Sphingosine-1-phosphate is higher in the cardiac phenotype of Fabry disease(only cardiac phenotype or better “in male patients with non-classic type of Fabry disease”? If I’m not wrong, there are 3 non-classic patients without HCM. Are there’s S1P values lower than the rest of non-classic patients with HCM?)

Response: We thank the reviewer for this comment and modified the title of our manuscript, which improves our manuscript. “Sphingosine-1-phosphate is higher in male patients with non-classic Fabry disease.”

 That is right, 3 early-diagnosed non-classic patients do not have HCM despite elevated S1P (268.2, 219.2 and 191.4 ng/ml). We thank the reviewer for this comment as it strengthens our message. We have added the following text to clarify (lines 99-100): Of notes, 3 early diagnosed non-classic patients without HCM had elevated S1P levels (268.2, 219.2 and 191.4 ng/ml).

  1. S1P appears to be positively correlated with non-classic type of Fabry disease. With respect to the patients’ characteristics, I would like to see the different genetic variants and the residual enzyme activity for missense mutations. I think that the classification in classic versus non-classic type will be more accurate using the residual enzyme activity.

Response:

In order to improve our manuscript, we provide a supplementary table with genetic variants and patient characteristics. Our cohort gathers patients from multiple centers with different diagnostic tools. Some patients only had the GLA gene variant identification. In those with available enzymatic activity, the method differed from a lab to another, and a consistent comparison might be challenging. Also, many patients were diagnosed earlier and have been treated for years, making the assessment of enzymatic activity difficult. We used the clinical classification that was previously reported (Mauhin et al, Plos One 2020), using the existence of cornea verticillata and the existence of acroparesthesia in males. This method showed the best results to identify clusters of patients in the absence of available residual enzyme activity.

  1. In the present cohort of non-classic FD the majority have HCM (except 3).

Patients in the non-classic type of FD are older, and age is directly correlated with HCM.

Response:

That is true. We think that non-classic FD patients are mainly diagnosed by cardiologists in front of HCM nowadays. Also, classic FD patients are younger in our cohort and most of them are treated, which can limit the development of HCM in this group. Because HCM is directly correlated to the age, we performed multivariate analysis to identify independent variables. Regarding patients heterogeneity limitation, a text has been added (lines 221-222): “The major limitation of this study is the heterogeneity of patients in terms of age and the prevalence of cardiac disease between phenotype groups. However, this limitation is inherent in studies in rare diseases”.

  1. Increase of IST may appear before HCM (what are the diagnostic criteria for HCM?)

Response : In the Material and Methods section we proposed that HCM would be defined if IST > 12 mm using cardiac echography, or cardiac MRI (line 241).

  1. Lyso-Gb3 is negatively correlated with the non-classic FD. Thus, apparently, S1P follows a different pathway as lyso-Gb3. Could you explain the relationship of S1P with alfa-Gal deficiency?

Could S1P elevation be an epiphenomenon secondary to lysosomal dysfunction or the inflammatory cascade? Is there any alteration of S1PRs that could produce an imbalance of S1P?

Response:

We thank the reviewer for highlighting this important point. The difference in S1P levels seems to be non-correlated with HCM or age, but with the phenotype. In classic patients, Aerts et al (PNAS 2008) demonstrated the pathogenic role of lysoGb3 on the smooth muscle cells of vessels with a proliferative effect. We now know that lysoGb3 levels are not highly elevated in non-classic patients despite the development of a similar cardiomyopathy. Brakch et al (Eur Heart J, 2010) demonstrated that mice treated with S1P developed a cardiac hypertrophy like FD. Hence, we can speculate that S1P plays the central role in the pathophysiology of non-classic FD. S1P mainly comes from sphingosine (through the action of sphingosine kinase), sphingosine comes from ceramide and ceramide comes from the catabolism of glycosphingolipids. The lack of alpha-galactosidase is supposed to decrease the downstream ceramide and therefore the sphingosine pool. However, we do not know the behavior of sphingosine kinase in sphingolipidosis and such imbalance between sphingosine and S1P has already been hypothesized in other sphingolipidoses such as acid sphingomyelinase deficiency (Mauhin et al, JCM 2021). We cannot yet consistently describe why S1P is different between phenotypes. This is a preliminary work that paves the way to further fundamental studies.

We tried to clarify this point in the discussion by mentionning the experiments of Brakch et al and Aerts et al. that demonstrated the proliferative effects of both compounds (lines 182-189): “Aerts et al. demonstrated the pathogenic role of lysoGb3 on the smooth muscle cells of vessels with a proliferative effect [11]. It is now admitted that plasma lysoGb3 is significantly lower in non-classic Fabry patients and even normal in some women [12]. Brakch et al. demonstrated that S1P also has a proliferative effect and mice treated with S1P developed a Fabry-like hypertrophic cardiomyopathy [3]. The results of these two experiments may explain the development of the Fabry cardiopathy in we consider that lysoGb3 and S1P play the major role respectively in the classic and non-classic phenotype.”

Aerts, J.M.; Groener, J.E.; Kuiper, S.; Donker-Koopman, W.E.; Strijland, A.; Ottenhoff, R.; van Roomen, C.; Mirzaian, M.; Wijburg, F.A.; Linthorst, G.E.; et al. Elevated Globotriaosylsphingosine Is a Hallmark of Fabry Disease. Proc. Natl. Acad. Sci. U. S. A. 2008, 105, 2812–2817, doi:10.1073/pnas.0712309105.

Brakch, N.; Dormond, O.; Bekri, S.; Golshayan, D.; Correvon, M.; Mazzolai, L.; Steinmann, B.; Barbey, F. Evidence for a Role of Sphingosine-1 Phosphate in Cardiovascular Remodelling in Fabry Disease. Eur. Heart J. 2010, 31, 67–76, doi:10.1093/eurheartj/ehp387.

Round 2

Reviewer 1 Report

We thank the authors for their answers. However  the study has several flaws that cannot be addressed without major methodological changes. In particular, we still do not understand why the late onset forms presents higher S1P levels than the classic forms yet their hypertrophic cardiomyopathy appears in mean ten years later.